# Versatility of Liposomes for Antisense Oligonucleotide Delivery: A Special Focus on Various Therapeutic Areas

**DOI:** 10.3390/pharmaceutics15051435

**Published:** 2023-05-08

**Authors:** Raghav Gupta, Sagar Salave, Dhwani Rana, Bharathi Karunakaran, Arun Butreddy, Derajram Benival, Nagavendra Kommineni

**Affiliations:** 1National Institute of Pharmaceutical Education and Research (NIPER), Ahmedabad 382355, India; 2Department of Pharmaceutics and Drug Delivery, School of Pharmacy, The University of Mississippi, Oxford, MS 38677, USA; 3Center for Biomedical Research, Population Council, New York, NY 10065, USA

**Keywords:** antisense oligonucleotide, liposomes, delivery system, cationic lipids, ASO generations

## Abstract

Nucleic acid therapeutics, specifically antisense oligonucleotides (ASOs), can effectively modulate gene expression and protein function, leading to long-lasting curative effects. The hydrophilic nature and large size of oligonucleotides present translational challenges, which have led to the exploration of various chemical modifications and delivery systems. The present review provides insights into the potential role of liposomes as a drug delivery system for ASOs. The potential benefits of liposomes as an ASO carrier, along with their method of preparation, characterization, routes of administration, and stability aspects, have been thoroughly discussed. A novel perspective in terms of therapeutic applications of liposomal ASO delivery in several diseases such as cancer, respiratory disease, ophthalmic delivery, infectious diseases, gastrointestinal disease, neuronal disorders, hematological malignancies, myotonic dystrophy, and neuronal disorders remains the major highlights of this review.

## 1. Introduction

Nucleic acid therapeutics have garnered considerable attention as an effective treatment modality for a vast array of inherited and acquired disorders. Precision and efficacy in the manipulation of cellular protein expression levels contribute to the enormous therapeutic potential of nucleic acid therapeutics [1]. Nucleic acid therapeutics can accomplish curative effects that are long-lasting by modulating gene expression via gene replacement, addition, inhibition, or editing, in contrast to conventional therapies, which show transient effects by targeting proteins [2]. Oligonucleotides are nucleic acid polymers composed of short, single- or double-stranded RNA or DNA molecules that comprise antisense oligonucleotides (ASO), RNA interference (RNAi), and aptamer RNAs. ASO and RNAi oligonucleotides are primarily used in the modulation of gene and protein expression, whereas aptamer oligonucleotides modulate the functions of proteins and other macromolecules by acting as “chemical antibodies” [3].

ASOs can be defined as single-stranded nucleic acid polymers (pieces of DNA) that are small (around 18–30 nucleotides), synthetic, composed of diverse chemistries, and complementary to the messenger RNA (mRNA) targets. They are employed for the modulation of gene expression via various mechanisms [4,5]. Upon binding of the DNA-based oligonucleotides to their cognate mRNA transcripts, the DNA-RNA heteroduplex substrates are formed, which further catalyzes the breakdown of RNA. Downregulation of the molecular target is the major goal of the antisense approach, which is often accomplished by inducing RNAse H endonuclease (one component of ASO) activity that recognizes and cleaves the RNA-DNA heteroduplex substrates. Destruction of the target RNA due to cleavage at the site of ASO binding leads to the silencing of target gene expression and diminution of target gene translation. The other potential mechanisms for the activity of ASO include inhibition of 5′ cap formation, modification in the process of splicing (splice-switching), and steric hindrance of ribosomal activity [4,6]. Briefly, ASO binds to mRNA and prevents its ability to make proteins, thus inhibiting translation and obstructing the production of faulty proteins. Functional mechanisms of ASOs are demonstrated in Figure 1. Upon entry into the nucleus, ASO binds to pre-mRNA and inhibits events related to the formation of the 5′ cap, splicing, RNA binding proteins, RNAse H1-mediated target degradation, translation, microRNA, and natural antisense transcripts. Table 1 represents approved ASOs in the clinic.

**Table 1 pharmaceutics-15-01435-t001:** List of approved ASOs in the clinic [7,8,9,10,11,12,13,14,15,16].

Name of ASO/ Company	Chemical Class	Target Organ	Route	Mechanism of Action	Indication	Approval Year
Vitravene (Fomivirsen) Ionis Pharma, Novartis	Phosphorothioate oligonucleotide	Eye	Intravitreal	Inhibit the replication of human CMV	Cytomegalovirus Retinitis	1998
Kynamro (Mipomersen) Sanofi/Isis	20-mer synthetic second-generation 2′-methoxyethyl antisense oligonucleotide	Liver	Subcutaneous	Hamper translation of ApoB-100 mRNA	Homozygous familial hypercholesterolemia	2013
Spinraza (Nusinersen) Biogen/Ionis	2′-O-(2-methoxyethyl) phosphorothioate antisense oligonucleotide	Central Nervous System	Intrathecal	Inhibits the SMN2 pre-RNA splicing	Spinal muscular atrophy	2016
Tegsedi (Inotersen) Ionis	2′-O (2-methoxyethyl)-modified phosphorothioate antisense oligonucleotide (ASO)	Liver	Subcutaneous	Degradation of wild-type and mutant TTR mRNA.	Hereditary-transthyretin mediated amyloidosis	2018
Waylivra (Volanesorsen) Ionis	2′-O-2-methoxyethyl (2′-MOE) antisense oligonucleotide (ASO)	Liver	Subcutaneous	Inhibits apo-lipoprotein C-III	Familial Chylomicronaemia	2019
Amondys 45 (Casimersen) Sarepta therapeutics	Phosphorodiamidate morpholino oligomer (PMO)	Muscle	Intravenous	Bind to exon 45 of the Duchenne muscular dystrophy (DMD) pre-mRNA and prevent translation	DMD	2021
Viltolarsen (Viltepso) NS Pharma	Phosphorodiamidate morpholino oligomer (PMO)	Muscle	Intravenous	Exclude the exon via binding to exon 53 of dystrophin pre-mRNA	DMD	2020
Exondys 51 (Eteplirsen) Sarepta therapeutics	Phosphorodiamidate morpholino oligomer (PMO)	Muscle	Intravenous	Alteration in exon splicing by binding to dystrophin	DMD	2016
Golodirsen (Vyondys 53) Sarepta therapeutics	Phosphorodiamidate morpholino oligomer (PMO) subclass	Muscle	Intravenous	Induce exon splicing	DMD	2019

## 2. Synthesis and Generations of ASOs

Generally, ASOs are synthesized by solid-phase synthesis, and based on structural modifications, ASOs can be categorized into the following types: first-generation ASO (backbone modified oligonucleotides), second-generation ASO (2′sugar–′modified oligonucleotides), and third-generation ASO (zwitterionic oligonucleotides) [18]. Figure 2 demonstrates the overall synthesis and purification methods for ASOs.

### 2.1. First-Generation ASO

First-generation ASO involves structural modifications such as 5′-N-carbamate, methylene-methylamine, amide, triazole, phosphorothioate, phosphorodithioate, thioether, thioformacetal, methylphosphonate, mercaptoacetamide, boranophosphate, N-3′-phosphoramidate, S-methylthiourea, and guanidinium, with the aim of overcoming limitations associated with conventional ASO [19]. Phosphorothioate oligodeoxynucleotides are the major representatives of this group. In the phosphorothioates, the non-bridging oxygen of the phosphate ester is replaced by a sulfur atom [20]. This simple modification has led to various favorable pharmacokinetic properties for the ASOs, including enhanced distribution into body tissues, a reduction in urinary excretion, and a prolongation of the residence time in tissues and cells because of significantly higher plasma protein binding [21]. Fomivirsene (Vitravene™), structurally (P-thio)-G-C-G-T-T-T-G-C-T-C-T-T-C-T-T-C-T-TG-C-G-desoxy-ribonucleic acid, is the first antisense drug developed and was approved by the FDA in 1998 for the management of cytomegalovirus-induced retinitis in patients with an acquired immune deficiency syndrome [18].

### 2.2. Second-Generation ASO

Second-generation ASO involves modification of the sugar moiety, specifically at the 2′ position, which includes the introduction of 2′-O-methoxyethyl and 2′-O-methyl groups. This modification leads to a desirable alteration in the pharmacokinetic, pharmacodynamic, and safety properties of ASO [22]. Significant improvement in binding affinity and resistance against enzymatic degradation has been demonstrated when the 2′ position was modified using ethylene glycol-based 2′-substituents (R = (CH2CHR′O) nOR″) [23].

Other modifications such as ASO containing 2′-O-(aminopropyl) cytidine, 2′-O-aminopropyl-substituted RNA (2′-O-(aminopropyl) adenosine, 2′-O (aminopropyl) guanosine, and 2′-O-(aminopropyl) uridine) were prepared by Griffey and his colleagues and demonstrated significant exonuclease resistance. This might be because of electrostatic repulsive forces operating between the side chain and the functional groups located at the enzyme’s catalytic site, which should dehydrate the phosphate group and subsequently bring in a catalyst with a positive charge such as a charged amino acid side chain or a hydrated metal ion before cleavage [24].

### 2.3. Third-Generation ASO

Third-generation ASOs are considered superior in comparison to first- and second-generation ASOs in terms of cell penetration, binding affinity, efficacy, resistance against nucleases, and negligible off-target effects. It involves chemical modification, including nucleobase modifications and bridged nucleic acids [25]. A specific type of nucleic acid known as “locked” nucleic acids is characterized by the presence of a methylene bridge, which connects the 2′ oxygen and 4′ carbon atoms of the ribose ring. The presence of a methylene bridge bond has been found to lock the flexibility of parent sugar, which further enhances the strength of its target interactions [26]. Miravirsen, a third-generation ASO, is one of the first miRNA therapies to be developed and undergo clinical testing. This ASO acts by targeting miR-122 for the management of hepatitis C infection [27]. The properties and chemical characteristics of different generations of ASOs are presented in Figure 3.

## 3. Interaction between ASO and Cell Membrane

In previous reports on the cellular uptake kinetics of S-oligos (phosphorothioate oligos) and based on the fact that S-oligos tend to compete with D-oligos (antisense deoxyoligonucleotides) for intracellular uptake, it can be inferred that these analogs may also gain entry into the cells through endocytosis, which could presumably be mediated through the same receptor protein [28,29]. MP-oligos (methyl phosphonate) are relatively hydrophobic and neutral in nature and were initially thought to gain entry into the cells via passive diffusion [30]. Akhtar et al. predicted through living cell models that simple diffusion is not likely to be the predominant mode of intracellular uptake for methyl phosphonate oligonucleotides. Further, authors have also suggested that, subsequent to endocytosis, the passage of oligonucleotides across the endosomal membranes has to be mediated by some mechanism other than passive diffusion [31]. Moreover, the information obtained from such studies is clearly not interpretable. Therefore, liposomes have been widely utilized to study the membrane permeation characteristics of small molecules.

Studies were conducted by researchers to evaluate the nature of oligonucleotide interaction with phospholipid membranes and to further study the permeation characteristics of oligonucleotide analogs across such membranes. It can be inferred from their findings that methyl phosphonates and, to a lesser extent, the analogs of phosphorothioate bind effectively to model lipid membranes. The binding characteristics of MP-oligos suggested the involvement of a simple molecular interaction in which each oligonucleotide molecule interacts with one binding site present on the lipid membrane [31].

## 4. Stability of ASOs

ASO-mediated therapies have been proven to be a more efficacious treatment option for various fatal conditions, such as spinal muscular atrophy and Duchenne muscular dystrophy, in comparison to available conventional treatment options [32]. However, the clinical application of ASO is limited because of their significantly higher susceptibility to degradation in vitro as well as in vivo by a variety of endo- and exonucleases. Several investigations have reported that single-stranded oligonucleotides are more susceptible to hydrolysis in comparison to double-stranded oligonucleotides. The hydrolytic activity is limited to the 3′ and 5′ positions, and it has been found that 3′ pyrimidine nucleotides were cleaved more rapidly than 3′ purines. In order to inhibit the rapid nuclease-mediated degradation of ASO in vitro and in vivo, modifications are required to be made at these 3′ and 5′ positions [33].

The other major obstacles associated with ASO-based therapies are poor intracellular delivery to the target site because of their inability to cross biological membranes, the potential for “off-target” effects, immunostimulatory activity, and other side effects [34]. The transport of the ASO from the circulation into the interstitium and then into the target cell (nucleus) to reach the target mRNA is required for successful gene silencing following systemic delivery. However, ASO may be degraded in the blood, removed by renal excretion, or taken up by tissue macrophages. Given their potential inability to traverse the vascular wall, ASO may be unable to reach their intended cell targets. ASO may be unable to reach their intracellular target (mRNA) once they have entered the interstitium due to their failure to either pass the cell membrane or escape the endosome-lysosome route [35]. The fate of ASO upon systemic administration is represented in Figure 4.

Lyophilization is a widely utilized technique for enhancing the stability of ASO-containing preparations. Studies have reported that lyophilized oligonucleotides remain stable for more than 3 years in general under refrigerated and frozen conditions [36]. While preparing ASO-containing liposomes, lyophilization was found to be a much more efficient method in comparison to rotary evaporation to remove the traces of any organic solvent that can hinder the formation of liposomes [37]. It has been observed that lyophilization can cause liposome fusion and phase separation during drying and rehydration. In order to overcome this limitation, lyoprotectants are used. Cryo/lyoprotectants limit mechanical damage and rupture of the lipid bilayer caused by ice crystals during the freeze-drying and rehydration processes by maintaining the membrane in a flexible state [38]. Meuiisnner and his coworkers developed immunoliposomes consisting of a core of ASO complexed with either a cationic lipid, 1,2-dioleoyl-3-trimethylammonium-propane, or a synthetic polycation, polyethyleneimine, encapsulated within liposomes containing a polyethylene-glycol derivative of distearoyl phosphatidylethanolamine. The prepared liposomes were lyophilized, and it was observed that freeze-dried liposomes can effectively protect the encapsulated nucleic acid for at least one year of storage [39]. Similarly, Zhong et al. developed a facile and reproducible method for large-scale preparation of MK-ASODN (midkine ASO) nanoliposomes followed by lyophilization successfully [40].

## 5. Route of Administration

Even though the first oligonucleotide-based therapy approved by the FDA, i.e., Vitravene™, utilized the intravitreal route of administration for the treatment of cytomegalovirus (CMV), various other routes for administration of ASO have been explored recently by researchers [41]. To effectively overcome the stability-related issues associated with ASO, liposomes have been utilized as nanocarriers for oligonucleotide delivery [42]. ASO-loaded liposomes have been administered via various routes for the treatment of a wide variety of diseases and infections.

Liu et al. intravenously administered antibody-modified liposomes with anti-miR-1 ASO (AMO-1) in mice to study their effect in the treatment of ischemic arrhythmia [43]. Similarly, ASO (bcl-2 AON sequence (5′-TCTCCCAGCGTGCGCCAT-3′) loaded poly (alkylene oxide)–poly (propyl acrylic acid) graft copolymers conjugated with cationic liposomes were administered via the intraperitoneal route in an animal model of ovarian carcinoma by Peddada and his colleagues [44]. In another study, antisense (myb-as) oligonucleotides encapsulated into liposomes coated with anti-GD2 (anti-disialoganglioside) antibodies were intravenously injected in a murine model of human neuroblastoma to study their efficacy in the treatment of neuroblastoma [45]. Wyrozumska and his colleagues prepared liposome-coated lipoplexes (L-cL) that consisted of a core resulting from the complexation of negatively charged oligodeoxynucleotides and positively charged lipids and studied them as a potential alternative to cancer therapy. In vivo studies were performed by injecting the nanocarrier via an intravenous route (tail vein) into the developed murine tumor model [46]. Similarly, in another study, liposomes encapsulated with the cyclopeptide RA-V (deoxybouvardin) and ASO (RX-004) were injected via tail vein into the mice to study the effect of this combination therapy in the treatment of hypoxic tumors [47].

A phase 1 clinical trial was carried out by Rudin et al. to study the effect of intravenously infused liposomes encapsulated with c-raf 1 ASOs for the treatment of advanced solid tumors [48]. Similarly, in another phase I clinical trial conducted in patients with advanced-stage cancer, Dritschilo and his colleagues evaluated the efficacy of liposomes encapsulated with c-raf antisense oligodeoxyribonucleotide when combined with radiation therapy. The drug was administered through the i.v. route 2 h before radiation therapy [49]. Garbuzenko and his colleagues comparatively studied the intratracheal and intravenous routes of drug administration for the delivery of ASO-loaded liposomes for the treatment of lung cancer. P-ethoxy-ASO was used as a payload for neutral PEGylated liposomes. The study results demonstrated that local intratracheal delivery of liposomal ASO for the management of lung diseases is more efficient in comparison to administering it by a systemic route [50].

## 6. Approaches for ASO Delivery

Several colloidal carrier systems have been explored for ASO delivery in various disease conditions, which are demonstrated in Figure 5. For instance, core-shell nanoparticles comprising of α-tocopherol succinate and poly (lactic acid)-g-poly (ethylene glycol) have been extensively researched for ASO delivery in lung cancer [51]. MALAT1-specific ASO and TAT peptide co-functionalized gold nanoparticles, namely ASO-Au-TAT NPs, that are capable of targeting the nucleus, were developed by Gong et al. The prepared nanoparticles imparted stability to ASO, exhibited improved internalization into the nucleus, and demonstrated good biocompatibility [52]. Yang et al. explored exosome-mediated delivery of ASO, which is capable of targeting alpha-synuclein for ameliorating the pathophysiology in a mouse model of Parkinson’s disease [53]. Biomimetic vesicles such as apoptotic bodies derived from tumor cells have also been explored for delivering ASO across the blood-brain barrier [54].

The increasing interest in ASO due to its ability to bind specifically to a target mRNA and thereby inhibit protein expression necessitates the development of robust delivery strategies that enable it to reach disease-associated tissues [6]. Despite having tremendous potential, the successful delivery of oligonucleotide therapies presents significant translational challenges. The major barrier to effective therapy using ASO arises from the fact that the sites targeted by these agents at the molecular level lie either in the nucleus or in the cytoplasm. These hydrophilic polyanions, due to their large size (~4–10 kDa), do not readily cross the plasma membrane [4]. Owing to the poor cell membrane permeability of oligonucleotides, large quantities need to be concentrated at the exterior of the cell to attain at least modest concentrations at the target site. This may not essentially serve as a major concern during in vitro experiments; however, it definitely proves to be a crucial issue in terms of its translational potential. Further, the lack of stability in the extracellular environment deteriorates its duration of action and potency [55]. Reticuloendothelial system uptake, renal clearance, degradation by nucleases in the extracellular matrix, and the non-productive sequestration caused by various plasma proteins limit their bioavailability upon systemic administration. To exert action, the oligonucleotide must first cross the capillary endothelium of the desired target cells within a tissue/organ via transcellular or paracellular routes, traverse the plasma membrane, escape the end lysosomal system to avoid lysosomal degradation or re-export via exocytosis, and finally reach the specific intracellular site of action.

Several approaches have been demonstrated to enhance the delivery of oligonucleotides [4,56]. Chemical modifications, including nucleic acid backbone modification, modification of the ribose sugar moiety and the nucleobase, backbone modification, stereochemistry-based changes, and bridged nucleic acid modifications, are employed [57,58]. Bioconjugation, covalent conjugation to lipid molecules, conjugation with carbohydrates (N-acetyl galactosamine), antibody and aptamer conjugates, peptide conjugates, nanocarrier based delivery systems, lipoplex and lipid-based approaches, exosome-based therapeutics, spherical nucleic acid approaches, DNA nanostructures, and stimuli-responsive nanotechnology are some of the therapeutic platforms that are being explored for delivering oligonucleotide drugs to specific tissues or organs [4,59]. Though chemical modification of oligonucleotides, such as the replacement of an oxygen group of the phosphate-diester backbone with a sulfur or methyl group, leads to enhanced nuclease resistance and stability, it is limited by poor intracellular uptake and non-sequence-specific effects [55].

## 7. Liposomes as a Promising Approach for ASO Delivery

As previously stated, although being a promising technique, the delivery of ASO treatments to specified tissues and cellular uptake are significant difficulties and constraints. Colloidal delivery, namely liposomes, is one of the approaches utilized to address such issues. Liposomes increase ASO stability in bodily fluids, facilitate drug distribution, improve cellular uptake, and allow ASO to bypass the endocytic process [25]. Liposomes have been employed as a drug delivery system for a wide range of drugs [60,61,62,63,64,65,66]. Liposomes are made up of one or more concentric, closed, phospholipid bilayer membranes that enclose an aqueous compartment on the internal side. Hydrophobic agents tend to partition into and become part of the phospholipid bilayer, whereas highly polar and water-soluble substances become entrapped in the liposome’s interior aqueous compartment.

Nucleic acid drugs can be incorporated into the aqueous compartment of liposomes. Several attempts have been made to employ liposomes in the delivery of ASOs due to their potential to effectively protect ASOs against nuclease-mediated degradation and enhance ASOs cellular uptake efficiency [42]. The use of cationic, anionic, fusogenic, pH-sensitive, and immunoliposomes for the delivery of ASO is well-established in the literature. In the case of anionic liposomes, the entrapment efficiency of nucleic acid is largely dependent on the concentration of the anionic lipid incorporated. Lipid concentrations greater than 20 mol% have been found to decrease the lipid-nucleic acid interactions owing to enhanced repulsion between the anionic lipid and the oligonucleotides, thus resulting in low encapsulation. Further, the ionic strength of the hydration buffer also influences encapsulation. Increasing the ionic strength enhances the bilayer lamellarity, which reduces nucleic acid encapsulation. Hence, these factors need to be optimized to achieve good results. In a study, the delivery of ASO to p53 (also known as tumor protein p53), encapsulated in anionic liposomes composed of dioleoyl phosphatidylcholine with 12 mol% of the anionic lipid dioleoyl phosphatidylglycerol, effectively prevented the upregulation of p53 protein expression. An increase in p53 protein expression is associated with an alteration in levels of redox proteins that causes apoptosis in the hippocampal neurons. In an in vitro excitotoxicity model, delivery of p53 ASOs encapsulated in anionic liposomes increased neuronal survival to approximately 75%. Further, the encapsulated p53 ASO was found to be neuroprotective at 5 to 10-fold-lower concentrations when compared to the unencapsulated ones [67].

Cationic liposomes exhibit high loading efficiency due to their tendency to bind to negatively charged nucleic acids, thereby forming structures known as “lipoplexes”. These lipoplexes show high transfection efficiency through endocytosis-mediated cellular internalization, which enables them to reach the target site located in the nucleus/cytoplasm. It is believed that upon internalization, the lipoplexes undergo fusion with the endosomal membrane, which in turn leads to the displacement of the nucleotide from the complex into the cytoplasm [68]. Cationic liposomes have been found to enhance the cellular localization of 99mTc-radiolabeled ASOs in target tumor cells by 4 to 5 folds in comparison to unencapsulated nucleotides. This increase in localization of the radiolabeled ASO has made imaging in nuclear medicine possible, which enables effective differentiation of gene overexpression in diseased tissues [69]. Cationic liposomes grafted with the monoclonal antibody Trastuzumab have been explored for the delivery of ASOs to prostate cancer cells. These immunoliposomes are capable of targeting HER-2, which is specifically expressed in prostate cancer cells. Upon delivery of the ASO to the cancer cells, silencing of the TCTP protein, which plays a crucial role in castration resistance, takes place, thus making immunoliposomes a promising drug delivery strategy in the field of oncology [70].

As with lipoplexes, polyplexes are complexes of nucleic acids with polycationic polymers, and in recent years, a new generation of carriers, called lipopolyplexes, has emerged [71]. Lipopolyplex is a ternary nanocomplex composed of a cationic liposome, polycation, and nucleic acid [72]. Encapsulating polyplexes with a lipid bilayer and thus forming a lipopolyplex may be a strategy that provides prolonged stability after intravenous administration while minimizing the harmful effect on normal cells [73]. Lipopolyplexes’ modular architecture allows for additional surface modifications, such as the addition of targeting molecules such as antibodies and their fragments, peptides and proteins, aptamers, or sugar moieties, without interfering with the structure of the polyplexes located inside the vesicles or the mechanical properties of the lipid shell. Moreover, lipopolyplexes exhibit many advantageous properties as nucleic acid carrier systems, which allows for further improvement of their effectiveness and biocompatibility. This is due to the overlap of superior properties of liposomal systems (i.e., lower cytotoxicity, relatively high cellular uptake, and stability) and polyplexes (i.e., facilitated endosomal escape, and efficient condensation of nucleic acids) [71]. This approach of condensing ASOs using cationic polymers and loading them into liposomes is widely investigated for the delivery of ASOs. For instance, in one study, transferrin (an iron transport protein that is frequently overexpressed on leukemia cells) conjugated lipopolyplexes (LPs), carrying the antisense oligonucleotide G3139 were characterized in an athymic mouse K562 subcutaneous xenograft model and examined for Bcl-2 downregulation and tumor inhibitory activity [74].

Smart drug delivery systems, such as pH-sensitive liposomes, undergo a phase transition from the lamellar state to inverted micelles at an acidic pH that initiates endosomal membrane fusion. The consequent destabilization of endosomes releases the oligonucleotides into the cytoplasm. However, these liposomes do not undergo degradation at the physiological pH of 7.4 and are thereby found to remain stable in plasma. One major challenge in the fabrication of pH-sensitive drug delivery systems is to facilitate the release of components into the cytoplasm before they reach the lysosomes to undergo degradation. The key strategy to achieve this is to enable the fusion of the liposomal and endosomal membrane bilayers by bringing them close to each other. The lipids that possess a small head group area tend to adapt inverted phases (negative membrane curvature), whereas the lipids that possess a large head group area tend to form micelles (positive membrane curvature), eventually causing fusion of the two types of lipid layers. Induction of physicochemical changes, such as modification of the interaction forces that alter the molecular geometry, plays a major role in the fusion of the membranes. The incorporation of a compound with a high pKa or an ion-binding agent such as Ca^2+^ is a common strategy to promote fusion through the enhancement of ionic forces between the two layers [75]. Dioleoyl-phosphatidylethanolamine (DOPE), owing to its propensity to adopt inverted hexagonal phases (HII), has been widely investigated for the fabrication of pH-sensitive liposomes. Under physiological conditions, these liposomes can be stabilized into bilayers by incorporating acidic lipids such as oleic acid and cholesterylhemisuccinate (CHEMS). The acidic lipids possess a negative charge at physiological (neutral) pH that contributes to electrostatic repulsion between DOPE molecules, thus preventing the formation of inverted hexagonal phases. However, in an acidic environment, the negative charge on the acidic lipids becomes neutralized through protonation, destabilization, and reversion of the DOPE molecules to their inverted hexagonal phase. The formation of the non-bilayer phase (HII) is responsible for the leakage of liposomal contents, i.e., the release of the encapsulated oligonucleotides [76].

## 8. Evaluation of Liposomal ASO Formulation

The characterization of different properties of liposomes, including size, colloidal behavior, phase transitions, and polymorphism, enables a thorough understanding of highly relevant pharmaceutical aspects such as biodistribution, storage stability, stability in circulation, targeting ability, and the propensity to release the drug following membrane fusion. Particle size is a critical determinant of the in vivo clearance rate of a liposomal formulation. The rate of clearance of liposomes from the human body is found to be directly proportional to the particle size. It has been observed that multilamellar vesicles with a larger particle size undergo rapid clearance in comparison to small unilamellar vesicles [77]. The particle size of the developed liposomes can be extensively measured by the quasielastic light scattering method, also called photon correlation spectroscopy (PCS). This method analyzes the intensity of scattered light that depends on particle size by means of autocorrelation analysis [78]. Further, the physical stability of a liposomal formulation following the entrapment of oligonucleotides is determined by its colloidal properties, which are characteristic of the forces that exist on the surface of liposomes. Surface modifications by alteration of the lipid composition, covalent linkage of proteins such as sugar-binding lectins, or incorporation of various synthetic polymers can be achieved to stabilize the formulation [79].

Guan et al. investigated liposomes as a potential carrier of ASOs against the androgen receptor for prostate cancer therapy. These liposomes, functionalized with a tumor-homing and penetrating peptide called iRGD, lead to an increase in their accumulation in the target tissues. The choice of lipid composition played a crucial role in the attainment of a desirable encapsulation efficiency and in enhancing the stability of liposomes. Two lipids, namely polyethylene glycol (PEG) conjugated 1,2-distearoyl-sn-glycero-3-phosphoethanolamine (DSPE) and dioleoyl-3-trimethylammonium propane (DOTAP), were utilized to provide a positive interior that can aid in the encapsulation of ASOs. Further, cholesterol and dipalmitoyl phosphatidylcholine (DPPC) were incorporated to enhance stability and thereby prevent the premature release of cargo from the carrier. The liposomes possessed a mean hydrodynamic diameter of around 150 nm and exhibited a uniform particle size distribution with a polydispersity index of less than 0.1. A negative surface charge was observed with a zeta potential of −6.67 to −7.10 mV. The drug loading efficiency and encapsulation efficiency were found to be 55.37 ± 1.6% and 83.36 ± 4.3%, respectively. The iRGD peptide grafted on to the free end of PEG was found to appear as a thin hydrated film representing the outer layer, as confirmed by transmission electron microscopy imaging [80].

The liposomes, upon attaining the lipid chain melting-transition temperature (Tm), undergo a conversion from the solid gel phase to the “fluid” liquid crystalline phase. At this phase, the lipid acyl chains become disordered and easily undergo lateral diffusion across the membrane. Most of the natural membrane lipids, for example, egg phosphatidylcholine (EPC), remain fluid at physiological temperatures. In order to study these phase transitions, differential scanning calorimetry (DSC) is most widely used. Since the transition temperature is responsive to additives in the membrane bilayer—for example, cholesterol decreases Tm and enhances the stability of the bilayer against leakage—it is an ideal parameter that needs to be monitored to understand the lipid-drug interactions and to check the presence of any impurities or breakdown products [78]. DSC also enables an investigation of the physical stability of liposomes that tend to undergo fusion into larger aggregates during storage. The fusion of lipids is characterized by the appearance of a new endothermic phase transition peak. Alternately, fluorescence-based lipid-mixing assays can be used to assess physical stability. In this technique, a fluorophore with aqueous solubility and its corresponding fluorescence quencher or enhancer are encapsulated in the core of the liposome, and the subsequent changes in fluorescence intensity upon liposomal fusion are quantified by Forster resonance energy transfer [81].

The encapsulation efficiency of oligonucleotides can be quantified using analytical techniques such as field-flow fractionation, capillary electrophoresis, and high-performance liquid chromatography (HPLC). Conventionally, to estimate the encapsulation efficiency, the unbound fraction is separated by ultrafiltration or ultracentrifugation and quantified. Based on the total amount initially added and the amount unencapsulated, the percent drug encapsulation can be calculated. In an attempt to develop a strategy to overcome multidrug resistance, Lo et al. prepared liposomal ASOs for MDR1, MDR-associated protein-1 (MRP1), MRP2, and BCL-2/BCL-xL. The encapsulation efficiency estimated based on HPLC analysis was found to be 86.28 ± 3.50% [82]. Lastly, in vitro cell uptake and in vivo biodistribution studies are carried out to ensure effective delivery of the ASOs into the cytoplasm/nucleus for gene silencing.

## 9. Applications of Liposomal ASO Formulations

The current therapeutic applications of ASOs are vast and include the treatment of several diseases such as renal diseases, spinal muscular dystrophy, respiratory diseases, osteoarthritis, etc. The section below discusses in detail the various applications of liposome-encapsulated ASOs in therapeutic medicine.

### 9.1. Myotonic Dystrophy

Myotonic dystrophy, a common form of inherited neuromuscular disease, occurs due to modifications in the gene encoding a protein kinase known as myotonin-protein kinase, or DMPK. The potential of ASOs to block DMPK expression in a reversible manner has gained wide attention. In particular, ASOs that possess modified internucleotide linkages, such as the phosphorothioate oligomers, can be specifically targeted against myotonin mRNA with high efficiency. MIO1, an ASO that targets the starting region of the first codon of DMPK, was synthesized in an attempt to inhibit the expression of DMPK. A 75% reduction in gene expression of DMPK was achieved in vitro on exposure to K562 and HepG2 cells after 6 h. To further enhance the uptake of oligonucleotides into the cells, MIO1 was complexed with a lipofectin reagent composed of two cationic lipids. The resultant liposomal formulation demonstrated a decrease in DMPK expression only after 24 h, indicating improved biological stability and an improved lifetime of the oligonucleotides in comparison to the unencapsulated ones [83].

Bubble liposomes have a long history of gene delivery. It encapsulates the gases, which could disrupt the liposomes upon external actuation. This system is dependent on ultrasound activation to release its payload [84]. Koebis et al. explored ultrasound-mediated delivery of oligonucleotides in bubble liposomes for the amelioration of myotonia in a mice model of myotonic dystrophy. The major factor that leads to myotonia in myotonic dystrophy patients is the abnormal splicing of the chloride channel 1 gene (CLCN1). The targeted delivery of phosphorodiamidate morpholino oligonucleotide (PMO) to the muscles corrects this alternative splicing of CLCN1. The administration of oligonucleotides through bubble liposomes, a currently emerging gene-delivery tool, followed by ultrasound exposure demonstrated an increase in the effectiveness of oligonucleotide delivery into the skeletal muscles of mice, making it a promising therapeutic strategy for myotonic dystrophy [85].

### 9.2. Respiratory Diseases

The cytokines, signaling molecules, and transcription factors play a critical role in the allergic, inflammatory, and immune responses involved in respiratory diseases such as asthma. The ASOs directed against these receptors/molecules present a promising strategy for the treatment of such diseases. The early signaling events in the pathophysiology of asthma are mediated by the tyrosine kinase SYK, against which phosphorothioate-modified ASOs have been effectively constructed. To facilitate its delivery through aerosolization, the oligonucleotides were complexed with the cationic lipid (1,2-dioleoyl-3-trimethylammonium-propane) and loaded onto a neutral lipid carrier composed of DOPE. Upon administration of the aerosolized formulation to rats, a significant down-regulation in SYK mRNA-mediated protein expression was observed in the alveolar macrophages. Further, inhibition of inflammatory cell infiltration in the airways that was triggered upon exposure to antigens was also observed. This exhibits the potential of the developed system to serve as an effective therapeutic strategy in the treatment of inflammatory diseases [86]. In patients with serious lung disorders such as cystic fibrosis, pathogens such as *Pseudomonas aeruginosa* exhibit a high level of drug resistance that eventually results in lung deterioration and premature death. To overcome this limitation, Fillion et al. constructed ASOs encapsulated in negatively charged fluid liposomes called “fluidosomes”. Fluidosomes are liposomes composed of DPPC and dimiristoyl phosphatidylglycerol (DMPG), which have a low gel-liquid crystalline transition temperature [87]. These “fluidosomes” demonstrated an increased potential to cross the bacterial cell membrane and thereby resulted in effective downregulation of bacterial gene expression [88].

### 9.3. Myocardium Dysfunction

Antisense strategies have been explored in various vascular diseases. Gene therapy that makes use of the antisense strategy for restenosis post-angioplasty has been reported [89,90]. However, the application of this strategy to cardiac diseases, including myocardial infarction and cardiomyopathy, is little known. This is due to the lack of suitable delivery methods for antisense oligodeoxynucleotides in vivo. To overcome this issue, Aoki et al. examined the application of the HVJ (Hemagglutinating Virus of Japan)-liposome-mediated strategy for in vivo oligodeoxynucleotide transfer into the heart of adult rats. Further, the author has compared the effectiveness of HVJ-liposomes with direct in vivo transfer into the heart through needle injection. Results suggested that after in vivo transfer through direct injection as well as in vitro transfer, fluorescence rapidly disseminated within one day, whereas HVJ-liposome-mediated transfer resulted in sustained fluorescence localized in the nucleus for at least one week. Furthermore, fluorescence measurement also exhibited a significantly higher level in the myocardium transfected by HVJ-liposomes than in the ones transfected by the direct transfer method. These results indicated that this method is not only an efficacious method for antisense oligodeoxynucleotide delivery but also helps to prolong the half-life of antisense oligodeoxynucleotide. This suggests the usefulness of gene therapy in cardiac disease [91].

### 9.4. Hepatic Disorders

Liposomal antisense therapy is used as a promising strategy for treating various liver diseases. In order to enable the effective delivery of various ASO into the hepatocytes for the management of Hepatitis-B viral infection, liver-targeting liposomes co-modified with the ligand of the asialoglycoprotein receptor (ASGPR) were studied by Zhang et al. [92]. Studies were conducted to evaluate the transfection efficiency and antigen inhibition potency of ASO-encapsulated cationic liposomes. The study results reported a high transfection efficiency along with a significant antigen inhibition effect in primary rat hepatocytes and HepG2.2.15 cells, respectively. Membrane fusion and endocytosis were reported as the main pathways of ASO cellular uptake. Results demonstrated that the co-modified liver-targeting cationic liposomes can be used as an effective carrier to transfer ASO into hepatocytes that are infected with the hepatitis-B virus [92].

### 9.5. Treatment of Colitis

The interaction between CD40 and CD40L tends to play a crucial role in the pathogenesis of experimental colitis. Therefore, Arranz et al. have evaluated the mechanism of action of ASOs against CD40, which was formulated in amphoteric liposomes (nov038/CD40) and systemically administered. For this, the experimenters induced colitis in Balb/c mice with 2,4,6-trinitobenzene sulfonic acid (TNBS) and later treated them with the formulation. Expression of CD40 was analyzed using flow cytometry on different cell subsets, and neo-immunity under CD40 modulation was determined by an antigen challenge model. The authors have observed that nov038/CD40 administration caused inhibition in the development of TNBS colitis. The novel formulation was found to be potent as it was able to completely suppress colitis upon single administration, and it also significantly lowered T cell activation and the level of pro-inflammatory mediators in serum. The inhibition of CD40 occurred specifically. Moreover, nov038/CD40 did not significantly affect the number of B or Treg cells. Therefore, it has been observed that the developed formulation has potent anti-inflammatory characteristics without being immunosuppressive [93].

### 9.6. Antiviral Therapy

In the realm of antiviral therapy, ASOs have a significant amount of untapped potential. These antisense molecules, which possess an oligodeoxynucleotide structure, are found to be complementary to specific sequences of the viral DNA or mRNA and thus have the potential to inhibit viral multiplication [94,95]. ASOs have good potential as antiviral drugs following methods that can efficiently deliver them to infected tissues and appropriate cellular compartments [96]. An improvement in plasma half-life and bioavailability can be achieved by using a liposomal drug delivery system. Moreover, employing these colloidal carriers such as liposomes also ensures a reduction in the degradation by nucleases. Ropert et al. encapsulated ASO in pH-sensitive liposomes to achieve active transport of the entrapped ASO and avoid lysosomal degradation. The pH-sensitive liposomes in this study were prepared using DOPE, cholesterol, and oleic acid in a ratio of 10:2:5 using the reverse phase evaporation method. During endocytosis, the pH is dropped in the endosomal compartment, and the developed liposomes become destabilized at an acidic pH. Following destabilization and fusion with the endosomal membrane, the liposome is expected to deliver its contents into the cytoplasm. The encapsulated ASOs have been shown to specifically inhibit the proliferation of the Friend retrovirus in mouse fibroblasts. Thus, in this study, it was demonstrated that 15-mer oligodeoxyribonucleotide was effective against the Friend retrovirus only when encapsulated or modified in liposomes, and the non-modified oligomer was found to be inefficient. Later, Ropert et al. measured the intracellular stability of this 15-mer oligodeoxyribonucleotide when modified or encapsulated and observed that there existed a direct correlation between the intracellular stability of the oligonucleotides and their antiretroviral efficiency [97].

### 9.7. Ophthalmic Therapy

ASOs have great scope in the field of antiviral therapy, notably for the treatment of intraocular infections, which mostly affect the eye’s posterior segment [94]. ASOs have been delivered to the ophthalmic areas to treat viral infections. Vitravene^®^ (a 21-mer phosphorothioate oligonucleotide) is an ASO-based intravitreal injection for CMV retinitis [7,41]. However, while the introduction of phosphorothioate oligonucleotides improved cell penetration and stability, it also resulted in several non-antisense actions. Phosphorothioate oligonucleotides have been reported as having the capacity to bind to a vast number of proteins in a manner that is independent of their sequence, resulting in substantial adverse effects. The delivery of drugs with a short half-life through the intravitreal route requires repeated administration. This leads to increased risks of endophthalmitis (eye infection), lens damage, retinal detachment, and poor tolerance in terminal-stage patients. Liposomes are an intriguing approach for improving the efficacy and comfort of intravitreal administration. Liposomal systems that are delivered intravenously have the ability to both greatly improve drug half-life and reduce intraocular adverse effects [94].

Several studies have been conducted on the use of liposomal ASO delivery systems for ocular indications. For the first time, Bochot et al. investigated the potential of liposomes for the intravitreal delivery of phosphodiester oligonucleotides. They developed sterically stabilized liposomes by a freeze-thaw method using soybean phospholipid, cholesterol, and PEG-DSPE in a molar ratio of 64:30:6 along with oligonucleotide 5′-phosphorylated oligothymidylate (pdT16). The entrapment efficacy of the developed formulation by the freeze-thaw method was reported to be 17.7 ± 4.7%, corresponding to a concentration of pdT16 of 44.3 ± 11.8 μM. The encapsulated liposomes had a particle size of 150 nm. The findings of the release study reported an initial burst release corresponding to approximately 10% of the total amount of encapsulated oligonucleotide, followed by sustained release. The liposomes could deliver oligonucleotides in a sustained manner from the vitreous humor. There was a significant reduction in the distribution of ASO in other nonrelevant tissues of the eye to maintain oligonucleotide integrity in the vitreous humor for a prolonged period and also avoid the unwanted aptamer effects associated with the use of phosphorothioate oligomers [94]. This offers interesting prospects for delivering intact oligonucleotides to the eye in a controlled manner for the treatment of retinal diseases. Likewise, Deng et al. also investigated the safety and efficacy of interleukin-6 ASO liposomes in experimental animals for the prevention of cataracts [98].

### 9.8. Antibacterial Therapy

The use of antisense therapy to treat bacterial infections is a very alluring alternative to dealing with the issue of antibiotic resistance due to its high specificity of action and little risk to human gene expression [99]. However, the implementation and development of such therapeutics have been challenging owing to the difficulty of ASOs penetrating bacterial cells. Fillion et al. encapsulated plasmid DNA and ASOs in a fluid, negatively charged (anionic) liposome and evaluated the potential of liposome-encapsulated ASOs to penetrate the bacterial outer membrane and inhibit gene expression in bacteria. The fluid liposomes were prepared by the dehydration-rehydration vesicle method using synthetic phospholipids DPPC and DMPG at a molar ratio of 10:1 as well as cholesterol. The liposomes were able to encapsulate significant amounts of plasmid and oligonucleotide DNA using monovalent salt solutions under vacuum conditions. Additionally, this research raises the possibility that anionic fluid liposomes targeted at an essential gene start codon could be effectively delivered into bacteria, limit the gene’s expression, and ultimately eliminate bacterial strains that are resistant to traditional antibiotics [88].

Das et al. developed a novel thiocationic (OBEHYTOP) liposomal formulation of phosphorothioate ASOs showing inhibitory activity against *Mycobacterium tuberculosis* as measured by an in vitro BACTEC 460TB assay. The liposomes were prepared by combining the salt of OBEHYTOP with a 10:5:2 molar ratio of thiocationic lipid: titratable amphiphile: sterol, respectively. Oleic acid was used as the titratable amphiphile, whereas vitamin D3 was used as the sterol. From the results, it was observed that oligonucleotides alone or the control liposomal vesicles lacked an inhibitory effect on *M. tuberculosis*. The hydrophilic oligonucleotides are not expected to be able to traverse the waxy outer coat of the bacteria, whereas, in the case of the liposomal system, the uptake of the oligonucleotides is facilitated by the presence of the OBEHYTOP interacting with the sulfatides present on the waxy coat of *M. tuberculosis*. Thus, the liposomal formulation can be employed to avoid the development of multidrug resistance to currently available antimycobacterial therapies [100].

### 9.9. Anti-Parasite Therapy

Control of human parasites has been made possible by the development of vaccines. However, the high variability of parasitic antigens and poor immunological responses to parasites pose challenges in their development. The finding that selective control of protein synthesis is made feasible by double-strand formation between mRNA and complementary (antisense) nucleic acids has proposed a different strategy for altering a pathogen’s metabolism. The oligodeoxynucleotides have shown lethal effects in *Trypanosoma brucei* following 3 h of incubation. However, if intracellular parasites are dealt with as opposed to external parasites, the issue becomes much more complicated. For instance, an oligodeoxynucleotide must pass through three membranes to treat malaria, as opposed to just one in the case of *T. brucei*. A conceivable solution could be to use antibody-coated liposomes to target the oligodeoxynucleotides in the host cell and promote their uptake. In comparison to unmodified oligodeoxynucleotides, poly(l-lysine)-conjugated S-oligodeoxynucleotides would have a higher efficiency of uptake by the parasite, better binding to target RNA, and improved nuclease resistance [101].

### 9.10. Anticancer Therapy

The clinical utility of ASOs is constrained by rapid in vivo cleavage and ineffective cellular uptake. The liposomal formulation might improve intratumoral ASO administration and prevent in vivo degradation. The first clinical evaluation of this concept using a liposomal ASO complementary to the c-raf-1 protooncogene was carried out by Rudin et al. [48]. Several liposomal ASOs have been studied against various oncogenes in a variety of cancer cell types. Grb2 is an adaptor protein that can associate either directly or indirectly with ErbB2, an oncogene associated with poor survival in breast cancer patients. Bcl-2 is a protein whose increased expression is associated with increased resistance to apoptosis. In one study by Siwak et al., Grb2 and Bcl-2 antisense molecules targeted against the translation initiation site were used. P-ethoxy oligonucleotides were incorporated into the liposomes made up of DOPC at a ratio of 1:10 (lipid:ASO). Liposomal Grb2 ASO treatment decreased cell growth in EGFR-overexpressing breast cancer cells, and liposomal Bcl-2 ASO treatment resulted in a significant induction of apoptosis in 11 of 19 patient samples [102]. Rodríguez et al. developed immunoliposomes carrying an ASO directed against human breast cancer cells overexpressing p185/HER2. The ASO directed toward the translational start site of dihydrofolate reductase RNA was utilized [103]. In one study, Pakunlu et al. demonstrated the capability of PEGylated liposomes to penetrate directly into tumor cells after systemic administration in vivo and successfully deliver (cytoplasmic and nuclear) an encapsulated anticancer drug (doxorubicin, DOX) and ASO. Encapsulation of DOX and ASO into liposomes substantially increased their specific activity. The ability of DOX to induce apoptosis, leading to higher in vitro cytotoxicity and in vivo antitumor activity, was significantly improved [104].

The spleen, liver, and bone marrow are the major sites where lymphoma and leukemic diseases are manifested. Liposomes, due to their ability to become widely distributed in these organs, can deliver ASOs specific to the target protein and thereby inhibit the proliferation of leukemic and lymphoma cells in these organs. This makes liposomes an ideal delivery system for the management of hematological malignancies. Tari et al. conducted a study in which liposomes loaded with ASOs were selectively taken up by K562 leukemic cells. This was followed by a sequence-specific inhibition in the production of target proteins that eventually resulted in the inhibition of leukemic and lymphoma cell growth. Further, biodistribution studies also showcased a wide distribution of liposomes in the liver in leukemia [37]. The off-target immunostimulatory effects of oligonucleotides are the major factor that limits their clinical translation. To overcome this limitation, Yu et al. developed antibody-conjugated liposomes for targeted delivery of the ASO G3139 against Bcl-2 mRNA. In addition to abrogating the off-target immunostimulatory effects, the liposomes also enhanced fludarabine-induced apoptosis of the desired cancer cells. The extensive overexpression of Bcl-2 in chronic lymphocytic leukemia (CLL) has been known to cause an increase in resistance to conventional cancer therapeutics such as fludarabine. However, the developed liposomes upon conjugation with the CD20 antibody, rituximab, enabled selective uptake of G3139 into the CD20-positive chronic lymphocytic leukemic B cells through the process of endosomal compartmentalization. This eventually resulted in robust down-regulation of Bcl-2, which in turn enhanced fludarabine-induced cytotoxicity of the CLL B cells. Further, based on the in vivo studies carried out in a xenograft leukemic model, a significant enhancement in the therapeutic efficacy was observed following the administration of the liposomes [105].

ASO has been studied for the treatment of various conditions associated with the nervous system. Neuroblastoma is one of the most common extracranial solid tumors that occur in infants. This neuroendocrine tumor has some unique features, including a high frequency of metastasis at diagnosis and an early age of onset [106]. Conventional treatment options available for treating this malignancy are inefficient, so novel therapeutic interventions need to be explored. Brignole and his colleagues developed coated cationic liposomes (CCLs), which comprise a central core that is made up of cationic phospholipid bound to c-myb ASO and an outer shell composed of neutral lipids. Further, a monoclonal antibody specific for the neuroectoderm antigen disialoganglioside GD2 was covalently bound to the external surface. The study results exhibited high loading efficiency for the ASO into the anti-GD2-targeted CCLs. Significantly higher inhibition of cell proliferation was also reported in comparison to non-targeted formulations or free ASO [107]. Thus, it can be concluded that targeted CCLs can be used as a potential delivery system for ASO to treat cancer.

In a study by Zhang et al., the effectiveness of ASO therapy for the clinical management of a pediatric brain tumor termed diffuse intrinsic pontine glioma was investigated. The presence of a point mutation in one of the two genes that encode for histone H3.3, involving the replacement of lysine 27 with methionine (K27M), is known to cause a significant reduction in the trimethylation of histone proteins on K27. This acts as a driving event in the induction of gliomagenesis. The developed 2′-O-methoxyethyl phosphorothioate ASOs were found to cause a RNase H-mediated knockdown of H3-3A mRNA and H3.3K27M protein, ultimately resulting in the restoration of the trimethylation of histone proteins. The subsequent reduction in tumor growth and enhanced neural stem cell differentiation in oligodendrocytes and astrocytes serve as clinical proof-of-concept for the effectiveness of ASO therapy in diffuse intrinsic pontine gliomas [108]. CNS tumors such as high-grade gliomas are associated with a high degree of mortality and morbidity, necessitating the development of new approaches for effective tumor management. The transforming growth factor beta-2 (TGF-β2), a polypeptide cytokine found to be specifically overexpressed in malignant gliomas, plays a crucial role in the malignancy and progression of the tumor. It has therefore been widely investigated as a potential target for cancer therapy. AP 12009, a TGF-β2 antisense compound, was found to cause a significant reduction in TGF-β2 secretion from patient-derived malignant glioma cells. Further, a dose-dependent inhibition in cell proliferation and migration with a significant enhancement in the immune cell-mediated cytotoxic antitumor response was observed. The safety and efficacy of AP 12009 have also been established in clinical phase I/II open-label dose escalation studies, making it a promising therapeutic approach for brain tumor therapy [109]. However, as the blood-brain barrier is impermeable to ASOs, the local administration has been the favored mode of drug delivery in such studies. Convection-enhanced delivery directly into the tumor tissue was used for the administration of AP 12009 owing to its ability to bypass the blood-brain barrier. Intrathecal administration of ASOs can also be considered a potential alternative route for delivery to the central nervous system as it results in significant bioavailability enhancement. Further, it causes low systemic exposure, thereby eliminating a major source of toxicity. However, the invasive nature of this route limits its use [27]. To alleviate the burden of currently used invasive routes of administration, Min et al. developed a glucose-coated polymeric nanocarrier for the systemic brain delivery of antisense oligonucleotides. ASO delivery across the blood-brain barrier was achieved by using glycemic control as an external trigger. The developed glucose-coated nanocarriers were able to be bound to the glucose transporter-1 expressed on brain capillary endothelial cells, thereby enabling the attainment of brain tissue accumulation following 1 h of intravenous administration [110]. Table 2 summarizes the liposomal ASO formulations for various disease conditions with their lipid composition, particle size, and route of administration.

## 10. Limitations of Liposomes for ASO Delivery

When a nucleic acid-loaded liposome reaches its site of transfection, i.e., the target cell, it has to overcome certain barriers for successful transfection. These barriers include:(i)Binding of the liposome to the cell surface;(ii)Endocytosis-mediated entry of the liposome into the cells or direct traversing of the plasma membrane (e.g., via membrane fusion);(iii)Liposomal escape from the endosome;(iv)Dissociation of the liposome to release the nucleic acid payload;(v)Transport through the cytosol;(vi)Entry into the nucleus.

Cationic liposomes have been widely investigated for the delivery of nucleic acids, as they can partially or fully protect them from degradation by various nucleases present in serum and also provide specific delivery into the target sites. However, the positive charge of this cationic liposome-nucleic acid complex (lipoplex) leads to non-specific electrostatic interactions of the liposome with various serum components, including lipoproteins, resulting in a decrease in subsequent gene expression in vivo [112]. The first-pass organs of the lung and liver only show a significant level of ASO expression in vivo [113]. Cytotoxicity associated with cationic liposomes, especially at higher concentrations, also limits their use both in vivo and in vitro [114]. Systemic administration of cationic particles and complexes, such as lipoplexes, also leads to rapid uptake by macrophages [115]. This can be attributed to their large size and excessive positive charge, which result in their clearance into cells of the mononuclear phagocytic system within a few minutes of their systemic administration. This unfavorable pharmacokinetics of liposomes results in sequestration of the encapsulated agent with organs of the mononuclear phagocyte system, thereby affecting liver and spleen function. Further, intravenous administration of liposomes leads to opsonization, thereby causing deleterious alterations to the healthy cells that come into contact with them during circulation. Liposomes can also stimulate a wide array of immune responses depending on their physicochemical properties, such as lipid composition, size, surface charge, and pegylation. Numerous studies have demonstrated that the intracellular uptake of cationic liposomes correlates with the degree of apoptosis and the generation of reactive oxygen species. Following the uptake of liposomes by the phagocytic cells and the subsequent generation of reactive oxygen species, damage to numerous cell organelles, such as the mitochondria, has been reported to occur. A wide range of cells demonstrated cytotoxicity upon increasing the value of the zeta potential beyond 30 mV in a charge-dependent manner [116]. Cationic liposomes composed of DOTAP have been reported to induce hepatotoxicity and pro-inflammatory responses in the C57BL/6 mice model, where a 3–6 fold increase in liver enzymes was found in the serum in comparison to neutral and negatively charged liposomes. Further, a 10–24-fold enhancement in the levels of Th1 and Th17 cytokines was observed at just 2 h post-injection. Based on such alarming results, it is evident that understanding the toxicities of liposomes to immune cells and their effect on the regulation of inflammatory responses are critical factors in avoiding the toxic effects of liposomes and therefore enhancing therapeutic benefits [117].

Anionic or neutral liposomes have significantly higher stability in serum along with increased circulation time, but reduced uptake by target cells and low genetic material loading capacity make them inferior to positively charged liposomes in ASO delivery [112]. In the case of ophthalmic delivery of ASOs, the corneal epithelial barriers limit the delivery of therapeutically active agents to the desired ocular areas. Effective intraocular penetration of ASOs is not achieved when they are administered topically, and liposomal formulations do not improve the penetration of topically applied ASOs [111]. Berdugo et al. investigated the possibility that iontophoresis could facilitate the transport of ASO to the cornea of the rat eye that was aimed at the vascular endothelial growth factor (VEGF)-R2 receptor (KDR/Flk). Fluorescence (CY5)-labeled ASO in phosphate-buffered saline (20 mM) was locally administered to rat eyes, and their fate within the anterior segment was studied. The ASO administered using transcorneoscleral iontophoresis was able to penetrate all layers of the cornea and was found in the iris. A corneal neovascular response was elicited prior to the application of ASOs, and in corneas with neovascularization, ASOs were preferentially localized within the vascular endothelial cells of the stroma [118]. Thus, it can be concluded that the transport of ASOs to the anterior segment of the eye, including all layers of the cornea, is facilitated by iontophoresis. Iontophoresis of ASOs directed at VEGF-R2 could be exploited to build a particular antiangiogenic strategy for the treatment of corneal disorders.

## 11. Conclusions

The translational potential of ASO in clinical settings has been mainly hampered by inadequate target specificity and the resultant off-target side effects. However, ASO possesses the potential to act as a frontier in therapeutic technologies, especially for rare diseases, and this can be witnessed in the future over a significant period of time. The emergence of ASO therapeutic agents, which need to be delivered by drug delivery carriers, is inevitable due to the improved stability of ASO in such carriers, which in turn would enhance the cellular uptake, enabling it to reach the target site for specific inhibition of gene expression. The employment of robust delivery strategies to enable ASO to reach disease-associated tissues is an unmet need. Liposomes are an ideal drug delivery carrier for ASOs due to their biocompatibility and biodegradability, as well as their ability to protect ASOs from degradation and enhance their cellular uptake. Overall, liposomes represent a promising new avenue for ASO delivery.

## Figures and Tables

**Figure 1 pharmaceutics-15-01435-f001:**
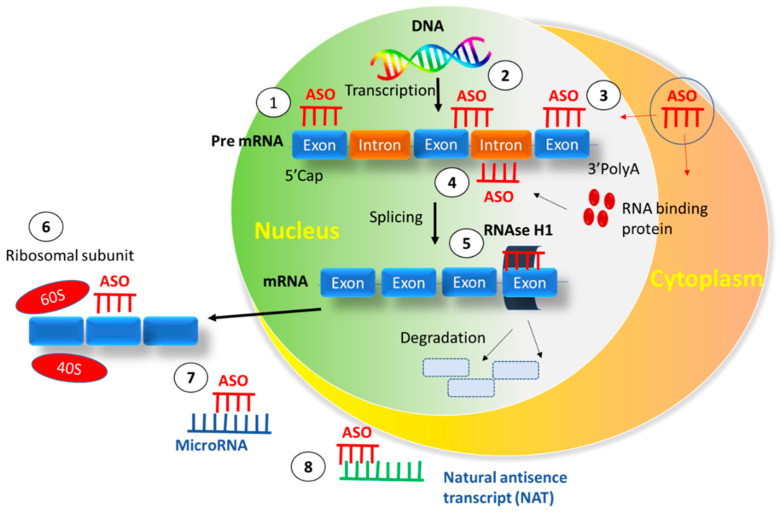
Functional mechanisms of ASOs involved in the regulation of targets. The number within the circle suggests the step-wise mechanism of ASOs during the regulation of targets. Reproduced with permission from reference [17].

**Figure 2 pharmaceutics-15-01435-f002:**
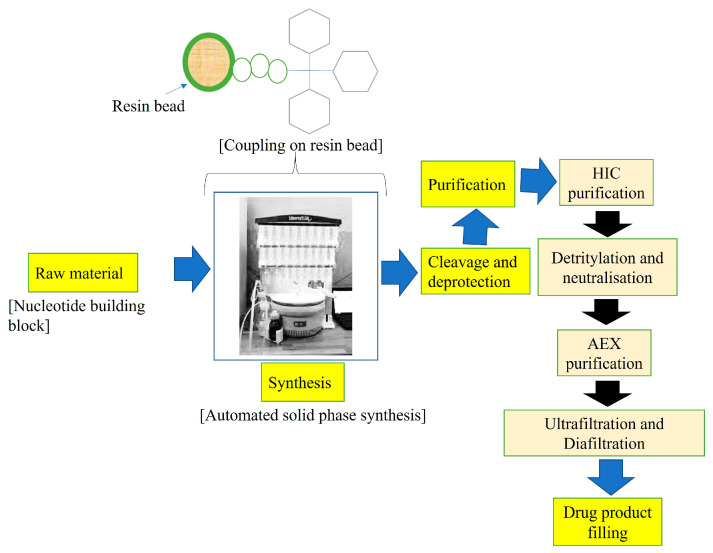
Overview of steps involved in ASO synthesis and purification.

**Figure 3 pharmaceutics-15-01435-f003:**
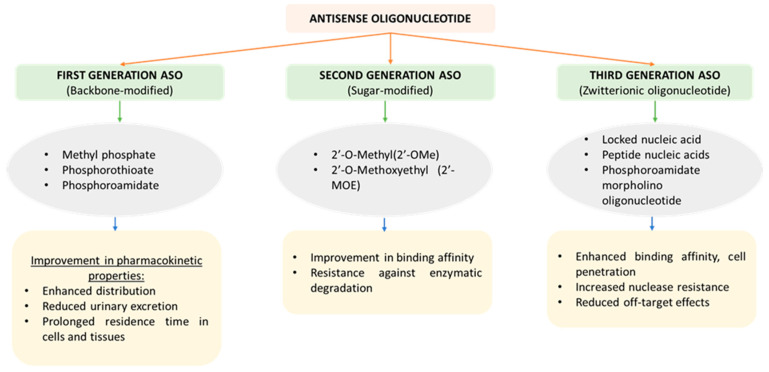
Schematic representation of different chemical characteristics of each generation of ASOs.

**Figure 4 pharmaceutics-15-01435-f004:**
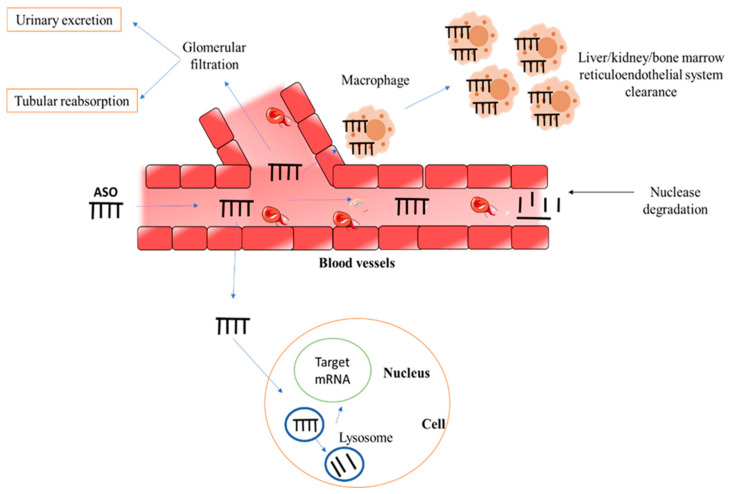
The fate of ASO upon systemic administration.

**Figure 5 pharmaceutics-15-01435-f005:**
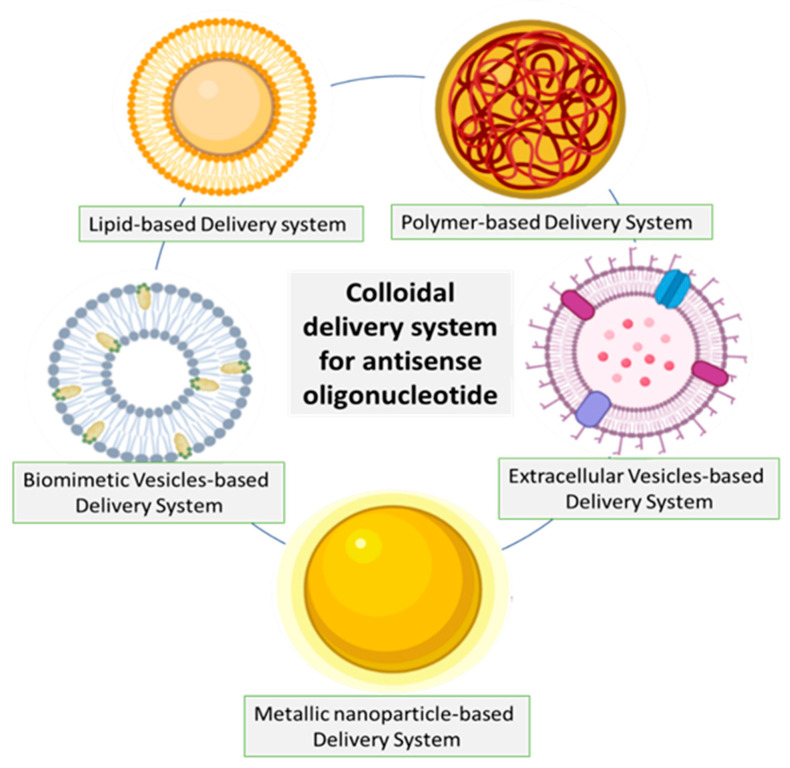
Reported colloidal systems for delivery of ASO.

**Table 2 pharmaceutics-15-01435-t002:** Liposomal ASO formulations.

Sr. No.	Disease	ASO	Lipid Composition	Particle Size	Route of Administration	Reference
1.	Chronic lymphocytic leuakemia	Phosphorothioate-modified oligos G3139 (5′-TCT CCC AGC GTG CGC CAT-3′), G3622 (5′-TAC CGC GTG CGA CCC TCT-3′), and a fluorescein-modified ODN (5′-(6)-FAM-TAC CGC GTG CGA CCC TCT-3′)	3β-N-(N′, N′-dimethyl amino ethane)-carbamoyl cholesterol/EPC/methoxy polyethylene glycol–distearoyl phosphatidylethanolamine (28/70/2)	56.3 ± 7.5 nm	Intraperitoneal	[105]
2.	Myotonic dystrophy	Phosphorodiamidate morpholino oligonucleotide (PMO)	DPPC:1,2-distearoyl-sn-glycero-3-phosphatidyl-ethanolamine-polyethyleneglycol (DSPE-PEG2000-OMe) (94:6)	<200 nm	Intramuscular	[85]
3.	Bacterial Infections	Anti-β-galactosidase antisense oligonucleotide (5′- GGT CAT AGC TGT TTC-3′)	DPPC: DMPG (10:1)	316.2–562.3 nm	-	[88]
4.	Neointimal hyperplasia	Antisense cdc2 kinase [5′-GTCTTCCATAGTTACTCA-3′]	Phosphatidylserine:Phosphatidylcholine:Cholesterol (1:4:8:2)	-	-	[89]
5.	Cancer	5′-CAG CGT GCG CCA TCC TTC CC-3′ and 5′-TTC AAG ATC CAT CCC GAC CTC GCG-3′ ASO	EPC/DPPC/Cholesterol (7:3:10)	100–200 nm	Subcutaneous	[104]
6.	Viral infections	5′-TGAACACGCCATGTC-3′ ASO	DOPE/Oleic acid/Cholesterol (10:5:2)	170 nm	-	[97]
7.	Ocular diseases	16-mer oligothymidylate (pdT16)	PC:Cholesterol:PEG-DSPE (64:30:6)	150 nm	Intravitreal	[111]
8.	Inflammatory Bowel Disease	Phosphorothioate and 2′methoxyethyl (MOE) modified ASO	POPC:DOPE:CHEMS:MoChol (15:45:20:20)	162 nm	Intravenous	[93]
9.	Cardiac diseases	FITC-labelled phosphorothioate ODN	Phosphatidylserine:Phosphatidylcholine:Cholesterol (1:4.8:2)	-	-	[91]
10.	Cardiac arrythmia	Anti-miR-1 ASOs (AMO-1)	EPC: CHO:DSPE-PEG2000 (49:50:1)	105 ± 0.5	Intravenous	[43]

## Data Availability

Not applicable.

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
