# Peer review of "Versatility of Liposomes for Antisense Oligonucleotide Delivery: A Special Focus on Various Therapeutic Areas"

_pharmaceutics, 2023, doi:10.3390/pharmaceutics15051435_

Round 1

Reviewer 1 Report

The manuscript is very interesting. It is substantially complete in addressing the various issues relating to the subject.

In the indications on their potential use, reported by the authors, there are no data relating to the use, if any, of ASOs in the treatment of brain tumours. I think a short section should be included in the text. The mechanisms by which ASOs combined with liposomes can overcome the blood-brain barrier are also not reported.

The possible toxicity of liposomes should also be discussed, even in the possibility of their prolonged use.

Author Response

Reviewer: 1

The manuscript is very interesting. It is substantially complete in addressing the various issues relating to the subject.

In the indications on their potential use, reported by the authors, there are no data relating to the use, if any, of ASOs in the treatment of brain tumors. I think a short section should be included in the text. The mechanisms by which ASOs combined with liposomes can overcome the blood-brain barrier are also not reported.

Ans: We appreciate the reviewer’s positive response to our work. As per the reviewer’s suggestion, the required information has been included in the revised manuscript and highlighted in yellow color.

The possible toxicity of liposomes should also be discussed, even in the possibility of their prolonged use.

Ans: As suggested by the reviewer, the toxicity of liposomes has been incorporated newly in the revised manuscript and highlighted in yellow color.

Reviewer 2 Report

There’s no doubt that antisense oligonucleotides (ASO) are valuable tools to broaden our perception of cell signaling and gene regulation. Growing mass of breakthrough discoveries in the area of genetics and therapeutic applications of various kinds of oligonucleotides give hope for developing new, highly effective drugs. In their manuscript entitled “Versatility of Liposomes for Antisense Oligonucleotide Delivery: A Special Focus on Various Therapeutic Areas” Gupta and collaborators attempt to highlight major discoveries in the field of ASO delivery carriers in light of mechanism of their action and major advantages of their use as potential therapeutics. The general flow of the text is smooth which makes it easily accessible to the reader, although it is a bit overgrown in regard to the content. Figures and tables further strengthen the story. Nevertheless, due to some issues listed below, the manuscript is premature and need further elaboration prior to its publication.

 - It is not clearly stated what is the selling point of the manuscript. This issue should be exposed in the text (particularly in the abstract), as there is a number of excellent reviews published very recently (including those cited within the manuscript, i.e. refs 4, 20, but also Nsairat et al. 2023 https://doi.org/10.1016/j.onano.2023.100132, to name a few) that overlap with the content of the submitted manuscript. 

- The authors should also consider fine-tuning the architecture of the manuscript. First of all, the volume of the text is not justified by the content, which is reflected by multiple repetitions (e.g. the issue of susceptibility of ASO to nucleases and protective role of liposomes in this regard). It is also hard to guess the logics behind sequential arrangements of chapters 3-6 (i.e. mixing information about ASO and ASO embedded in liposomes). Secondly, it is not straightforward what is the key the authors used while arranging chapters 8 and 9 (e.g. why the content subchapters 8.1 and 9.5 is split).

- Table 1 is to a large degree incomplete, as it does not contain all the ASOs approved to clinical use (e.g. Casimersen is missing). It would be also informative to classify all the items mentioned in the table according to the generation of ASOs (see chapter 2)

- I think that Figure 2 is redundant as it does not provide any information related to the main message of the manuscript.

- It its current form Figure 5 is also of limited utility. The authors should focus only on the aspects directly linked to liposomes used as ASO cariers (e.g. is it possible to perform active loading of ASOs into liposomes?). Moreover, the figure suggests that there’s only one “method of preparation” possible, i.e. “reverse-phase evaporation vesicles”.

- Table 2 should also include only to methods that refer to ASO-loaded liposomes and should obtain information at which step (in which phase) ASOs are added to the system.

- The authors should also consider clearer introduction of non-standard items mentioned in the text such as “bubble liposomes” “fluidosomes” etc. It is also confusing what iontophoresis has in common with liposomal formulations of ASOs – the authors should clarify it or remove the relevant fragment (l.729-743).

- One of the most widely used approaches is based on lipopolyplexes, where cationic polymers are used to condense ASOs and load them into liposomes. The authors should also include these into the discussion. In this regard, some recent reviews could be helpful, e.g. Jerzykiewicz & Czogalla 2022, Materials 15:179, Rezaee et al. 2016 J Control Release 236:1-14 (and literature cited therein).

- For clarity it is highly recommended to prepare a table summarizing all the liposomal formulations mentioned in chapters 8 and 9 in respect to: lipid composition, particle size, route of administration, tissue/cellular/molecular targets etc.

- Within conclusions the authors claim that “… liposomes serve as an ideal drug delivery carrier to overcome the aforementioned limitations of ASO delivery.”, but do not summarise what are their advantages over other ASO carriers (such as polyplexes, dendimers etc.). A reader expects some more substantive argumentation.  

- It is a bit surprising that the authors introduce some abbreviations (e.g. DOTAP, DOPE) and refuse to use them later. 

- Some minor issues include: no description what the numbers within Figure 1 state for; some conceptual errors (e.g. “…the delivery of ASO to p53…”); numerous errors within reference list (e.g. redundand elements in [20], missing journal names in [43] and [45]) and inconsistencies in nomenclature.

Author Response

Reviewer: 2

There’s no doubt that antisense oligonucleotides (ASO) are valuable tools to broaden our perception of cell signaling and gene regulation. Growing mass of breakthrough discoveries in the area of genetics and therapeutic applications of various kinds of oligonucleotides give hope for developing new, highly effective drugs. In their manuscript entitled “Versatility of Liposomes for Antisense Oligonucleotide Delivery: A Special Focus on Various Therapeutic Areas” Gupta and collaborators attempt to highlight major discoveries in the field of ASO delivery carriers in light of mechanism of their action and major advantages of their use as potential therapeutics. The general flow of the text is smooth which makes it easily accessible to the reader, although it is a bit overgrown in regard to the content. Figures and tables further strengthen the story. Nevertheless, due to some issues listed below, the manuscript is premature and need further elaboration prior to its publication.

 - It is not clearly stated what is the selling point of the manuscript. This issue should be exposed in the text (particularly in the abstract), as there is a number of excellent reviews published very recently (including those cited within the manuscript, i.e. refs 4, 20, but also Nsairat et al. 2023 https://doi.org/10.1016/j.onano.2023.100132, to name a few) that overlap with the content of the submitted manuscript.

Ans: Regarding the recent reviews cited in the manuscript and the Nsairat et al. (2023) paper you mentioned, we agree that they overlap with some of the content of our manuscript. However, our study provides a novel perspective in terms of therapeutic applications of liposomal ASO delivery in several diseases and additional insights relating to liposomal delivery that we believe are valuable to the field. Moreover, the revised abstract is highlighted in yellow.

- The authors should also consider fine-tuning the architecture of the manuscript. First of all, the volume of the text is not justified by the content, which is reflected by multiple repetitions (e.g. the issue of susceptibility of ASO to nucleases and protective role of liposomes in this regard). It is also hard to guess the logics behind sequential arrangements of chapters 3-6 (i.e. mixing information about ASO and ASO embedded in liposomes). Secondly, it is not straightforward what is the key the authors used while arranging chapters 8 and 9 (e.g., why the content subchapters 8.1 and 9.5 is split).

Ans: As per the reviewer’s suggestion, the whole content of the manuscript has been rearranged. Chapter 3-6 has been moved in earlier text and chapter 8 and 9  are merged.    

- Table 1 is to a large degree incomplete, as it does not contain all the ASOs approved to clinical use (e.g. Casimersen is missing). It would be also informative to classify all the items mentioned in the table according to the generation of ASOs (see chapter 2)

Ans: As per the reviewer’s suggestion, table 1 has been modified and included in the revised manuscript. 

- I think that Figure 2 is redundant as it does not provide any information related to the main message of the manuscript.

Ans: We agree with the reviewer’s comment that Figure 2 does not provide any information related to the main message of the manuscript. However, it demonstrates the overview of ASO synthesis which would be useful to the reader for better understanding.

- It its current form Figure 5 is also of limited utility. The authors should focus only on the aspects directly linked to liposomes used as ASO cariers (e.g. is it possible to perform active loading of ASOs into liposomes?). Moreover, the figure suggests that there’s only one “method of preparation” possible, i.e. “reverse-phase evaporation vesicles”.

 Ans: We appreciate the reviewer’s keen observation. Although several manufacturing methods are available for liposome preparation, reverse -phase evaporation method is used widely in liposomal carrier systems for ASO delivery. Further, the Passive loading of ASO into liposomes has been carried out by several researchers. Additionally, to align with reviewer 3 comment, we have excluded this figure from the revised manuscript.  

- Table 2 should also include only to methods that refer to ASO-loaded liposomes and should obtain information at which step (in which phase) ASOs are added to the system.

Ans: The film hydration method is used for the development of ASO-containing liposomes. In this method loading of ASO is take place during the hydration step. As per the reviewer’s comment, the table should contain methods that refer to ASO-loaded liposomes therefore due to a lack of suggested information, the author would like to exclude table 2 from the revised manuscript.   

- The authors should also consider clearer introduction of non-standard items mentioned in the text such as “bubble liposomes” “fluidosomes” etc. It is also confusing what iontophoresis has in common with liposomal formulations of ASOs – the authors should clarify it or remove the relevant fragment (l.729-743).

Ans: As per the reviewer’s suggestion, the required changes have been included in the revised manuscript and highlighted in yellow color.

Iontophoresis is not common in all ASO-based liposomal therapies. However, one of the researcher (Berdugo et al.) have evaluated the possibility of iontophoresis to facilitate the transport of ASO to the rat eye cornea.

- One of the most widely used approaches is based on lipopolyplexes, where cationic polymers are used to condense ASOs and load them into liposomes. The authors should also include these into the discussion. In this regard, some recent reviews could be helpful, e.g. Jerzykiewicz & Czogalla 2022, Materials 15:179, Rezaee et al. 2016 J Control Release 236:1-14 (and literature cited therein).

Ans: Lipopolyplex approach is an important strategy for delivering ASOs and as suggested by the reviewer, necessary changes have been made in the revised manuscript and highlighted in yellow color.

- For clarity it is highly recommended to prepare a table summarizing all the liposomal formulations mentioned in chapters 8 and 9 in respect to: lipid composition, particle size, route of administration, tissue/cellular/molecular targets etc.

Ans: As per the reviewer’s suggestion, the required table has been included in the revised manuscript and highlighted in yellow color.

- Within conclusions the authors claim that “… liposomes serve as an ideal drug delivery carrier to overcome the aforementioned limitations of ASO delivery.”, but do not summarize what are their advantages over other ASO carriers (such as polyplexes, dendrimers etc.). A reader expects some more substantive argumentation.  

Ans: The required information has been included in the revised manuscript and highlighted in yellow color.  

- It is a bit surprising that the authors introduce some abbreviations (e.g. DOTAP, DOPE) and refuse to use them later. 

Ans: As per the reviewer’s suggestion, abbreviations have been corrected in the revised manuscript.

- Some minor issues include: no description what the numbers within Figure 1 state for; some conceptual errors (e.g. “…the delivery of ASO to p53…”); numerous errors within reference list (e.g. redundand elements in [20], missing journal names in [43] and [45]) and inconsistencies in nomenclature.

Ans: As per the reviewer’s suggestion, minor issues have been rectified in the revised manuscript and the references have also been formatted as per the journal’s guidelines.

Reviewer 3 Report

This review article focuses on using antisense oligonucleotide therapy against different disorders. The authors provide a comprehensive overview of the evidence regarding the advantages of using liposomes to deliver these genetic materials. This review can be interesting for those working in the field and for a wider audience. However, the authors need to work more on the text to make it better focused and condensed. In its present form, there are numerous repeats, and the whole text of the paper is unnecessarily stretched, referring to the same ideas through different wording across many subsections. Some rearrangements of the text are also needed. For example, section 4 about the ‘interaction between ASO and cell membrane’ should be given earlier in the text. Similarly, for section 6, ‘stability of ASO’. The authors mentioned existing colloidal systems for the delivery of ASO (Figure 4) but did not explain why they focused on liposomes in their review.

Some other issues requiring attention to improve the general quality of the manuscript are as below:

- The abstract appeared somewhat chaotic and must be structured.

- A list of all abbreviations used across the text would help to navigate the readers. 

-  In figure 6, giving the approximate per cent of antisense oligonucleotides following systemic administration would make more clarity. 

Author Response

Reviewer: 3

This review article focuses on using antisense oligonucleotide therapy against different disorders. The authors provide a comprehensive overview of the evidence regarding the advantages of using liposomes to deliver these genetic materials. This review can be interesting for those working in the field and for a wider audience. However, the authors need to work more on the text to make it better focused and condensed. In its present form, there are numerous repeats, and the whole text of the paper is unnecessarily stretched, referring to the same ideas through different wording across many subsections. Some rearrangements of the text are also needed. For example, section 4 about the ‘interaction between ASO and cell membrane’ should be given earlier in the text. Similarly, for section 6, ‘stability of ASO’. The authors mentioned existing colloidal systems for the delivery of ASO (Figure 4) but did not explain why they focused on liposomes in their review.

Ans: As per the reviewer’s suggestion, the whole manuscript has been evaluated for content repetition. Further, suggested sections have been moved to earlier in the text, and the rationale for focusing liposomal formulation in ASO delivery has been included in the revised manuscript and highlighted in yellow color.

Some other issues requiring attention to improve the general quality of the manuscript are as below:

- The abstract appeared somewhat chaotic and must be structured.

Ans: As per the reviewer’s suggestion, the whole abstract has been restructured.  

- A list of all abbreviations used across the text would help to navigate the readers.

Ans: As suggested by the reviewer, a list of abbreviations has been included in the revised manuscript and highlighted in yellow color.   

-  In figure 6, giving the approximate per cent of antisense oligonucleotides following systemic administration would make more clarity. 

Ans: We appreciate the reviewer’s keen observation; However, we did not find any relevant literature for the same. 

Round 2

Reviewer 2 Report

I would like to thank the authors for addressing all the issues raised by the reviewers. The manuscript improved substantially and now it is ready for further processing and publication.

Reviewer 3 Report

The authors have addressed suggestions regarding improving the paper and the value of its content. I endorse the publication.